# Superhydrophobic and Electrochemical Performance of CF_2_-Modified g-C_3_N_4_/Graphene Composite Film Deposited by PECVD

**DOI:** 10.3390/nano12244387

**Published:** 2022-12-09

**Authors:** Dayu Li, Yuling Lu, Chao Zhang

**Affiliations:** School of Mechanical Engineering, Yangzhou University, Yangzhou 225009, China

**Keywords:** functional graphene, superhydrophobic films, electrochemical reaction, plasma processes, first principles calculation

## Abstract

The physicochemical properties of functional graphene are regulated by compositing with other nano-carbon materials or modifying functional groups on the surface through plasma processes. The functional graphene films with g-C_3_N_4_ and F-doped groups were produced by controlling the deposition steps and plasma gases via radio frequency plasma-enhanced chemical vapor deposition (RF-PECVD). The first principles calculation and electrochemistry characteristic of the functional graphene films were performed on Materials Studio software and an electrochemical workstation, respectively. It is found that the nanostructures of functional graphene films with g-C_3_N_4_ and F-doped groups were significantly transformed. The introduction of fluorine atoms led to severe deformation of the g-C_3_N_4_ nanostructure, which created gaps in the electrostatic potential of the graphene surface and provided channels for electron transport. The surface of the roving fabric substrate covered by pure graphene is hydrophilic with a static contact angle of 79.4°, but the surface is transformed to a hydrophobic state for the g-C_3_N_4_/graphene film with an increased static contact angle of 131.3° which is further improved to 156.2° for CF_2_-modified g-C_3_N_4_/graphene film exhibiting the stable superhydrophobic property. The resistance of the electron movement of CF_2_-modified g-C_3_N_4_/graphene film was reduced by 2% and 76.7%, respectively, compared with graphene and g-C_3_N_4_/graphene.

## 1. Introduction

Graphene with a two-dimensional nanostructure possesses unique optical, electrical, and mechanical properties, which helps it work as a popular functional material [1,2,3]. The main preparation methods for high-quality graphene are the micro-mechanical stripping method [4], the epitaxial growth method [5], the graphite oxide reduction method [6], and the chemical vapor deposition method [7]. The graphene prepared using the micromechanical exfoliation and epitaxial growth method presents as single-layer graphene with high purity [8]. However, the graphene prepared with such technology is only used at the laboratory level due to its extremely low output and high requirements for hardware facilities, which greatly affects the commercialization process of graphene. The reduced graphite oxide method (Hummer’s method) is the main method used to prepare graphene by mixing graphite powder and a strong oxidant and reducing graphene oxide with hydrazine [9]. The graphene prepared using Hummer’s method promotes a large output, however, the obtained graphene has such great defects that it partially loses its outstanding physicochemical properties; meanwhile, the highly toxic hydrazine hydrate may lead to great safety hazards in waste liquid treatment [10]. The graphene prepared using CVD has the characteristics of high controllability, scalability, and relatively low cost, and the synthesized graphene has good homogeneity, which makes it suitable for large-area preparation. Therefore, the research on the synthesis of graphene by CVD is always concerned with this process [11,12]. The graphene prepared by CVD has a certain theoretical basis to become a promising process, especially in commercial mass production. The physical and chemical properties of graphene are effectively regulated by compositing with other nano-carbon materials or changing the types of functional groups on the graphene surface, resulting in a product that is defined as functional graphene [13].

Many researchers have chosen graphite-phase carbon nitride (g-C_3_N_4_) with high conductivity and photocatalytic activity to construct graphene and carbon nitride nanocomposite photocatalytic materials, which can thus not only retain their excellent properties but also produce synergistic effects to improve electrochemical activity [14,15,16]. The C and N atoms in the structure of g-C_3_N_4_ form a highly delocalized π conjugated system with sp^2^-C hybridization, which exhibits a suitable semiconductor band edge position and meets the thermodynamic requirements of photo-hydrolysis of aquatic hydrogen to produce oxygen. Dong [17] investigated the structural and electronic properties of a g-C_3_N_4_/graphene/g-C_3_N_4_ sandwich heterostructure using density functional theory with van der Waals correction. The results showed that the band gap of the sandwich heterostructure could be as high as 106 meV without strain while maintaining high carrier mobility. Wu [18] studied the atomic structure and electronic properties of hybrid g-C_3_N_4_/graphene nanocomposites using first principles calculations. The calculation results showed that the buckle graphene/g-C_3_N_4_ nanocomposite was more stable than the planar one, and graphene as a substrate of graphene/g-C_3_N_4_ nanocomposite can obviously loosen the buckle degree of g-C_3_N_4_ to stabilize electron configuration. The main synthesis methods of g-C_3_N_4_ are mainly divided into solid phase reaction method, solvothermal method, electrochemical deposition method, and thermal polymerization [19,20,21,22]. Although film preparation by CVD is currently the mainstream process in the industrial production process, the study of g-C_3_N_4_ one-step preparation by CVD is rarely reported. In addition, the activity of nano-carbon-based materials can also be adjusted by doping atoms (such as F, B, N, P, S, etc.), so as to achieve the purpose of adjusting the charge distribution, electronegativity and electron transfer behavior [23,24]. The special C-F bond and lamellar structure of fluorine-doped nanocarbon materials give them special properties such as low surface energy, strong hydrophobicity, self-lubrication, corrosion resistance, and friction resistance [25,26]. Based on these excellent properties, there have a great number of applications in different fields containing superhydrophobic materials [27], thermoelectric and optoelectronic devices [28], and corrosion-resistant coatings [29]. Jayasinghe [30] used CF_4_ plasma to fluorinate multi-walled carbon nanotubes and applied it to lithium primary battery and lithium secondary battery and obtained slightly fluorinated multi-walled carbon nanotubes with improved cycling performance compared with the original carbon nanotubes.

Plasma technology, as a gaseous treatment technology that combines physical and chemical methods, has attracted more and more attention in material modification because of its advantages such as controllable operation, high treatment efficiency, mild reaction conditions, green environmental protection, and targeted modification of material surface without damage to the internal structure [31]. Within the field of plasma technology, PECVD (Plasma Enhanced Chemical Vapor Deposition) is a typical “bottom-up” deposition method. By ionizing the reaction gas to form a plasma, the active groups dissociated from the gas molecules after high-energy electron bombardment are deposited on the substrate surface to form the final product. Compared with the traditional thermal CVD method, the PECVD method can achieve high-quality controllable material preparation at a lower temperature without a catalyst, which has been extensively applied to realize the regulation of defects, the preparation of layered composite films, and the doping of heterogeneous atoms [32]. Yi [32] reviewed the research progress of two-dimensional materials prepared by the PECVD method and their application in recent years and found that there were different competition processes among etching, nucleation, and deposition in PECVD. The two-dimensional high-quality materials with different morphologies and properties prepared at different equilibrium states have broad application prospects in photodetectors, pressure sensors, biochemical sensors, electronic skin, Raman enhancement, etc. This “bottom-up” method can effectively prepare carbon nanomaterials, facilitate in situ heteroatom doping of carbon nanomaterials, modify the material surface functional groups or construct structural defects, and regulate the electronic structure to meet the needs of different applications [33,34,35]. Woon’s team [33,34] used low-power capacitance-coupled radiofrequency plasma-enhanced chemical vapor deposition to grow fully covered, mostly single-layer graphene films with serious defects on copper substrates by controlling the proportion of plasma gas sources. Additionally, it was found that the vacancy defects of the graphene films were caused by ion bombardment during growth, which caused the change of charge transfer in the graphene films.

In this work, the functional graphene films were produced by PECVD technology, and the related properties were controlled by tuning the plasma deposition steps and injecting different plasma gases. The novelty of this study is that the in-situ growth of CF_2_-modified g-C_3_N_4_/graphene films with graphene primer improves the bonding performance and durability of the superhydrophobic film and uses the wide band gap of g-C_3_N_4_ to improve the corrosion resistance of the composite film in the salt solution. Concurrently, the combined method of experiment and modeling is used to investigate the results, and the relationship between structures and properties is studied by building suitable models and detailed characterized by using different experimental techniques.

In this research, aimed to optimize the superhydrophobic and electrochemical performance of graphene film, the g-C_3_N_4_ and F-doped functional groups were mixed by controlling the deposition steps and plasma gases using PECVD technology. The related micro-nano structures of the produced films were determined by TEM, XPS, and FTIR, and their atomic models were established through Material Studio 7.0. The crystal model was established by density functional theory calculation.

## 2. Materials and Methods

### 2.1. Materials

The Ar/CH_4_ mixed gas (CH_4_ concentration: 5.01 × 10^−2^ mol/mol), Ar/NH_3_ mixed gas (NH_3_ concentration: 1.02 × 10^−4^ mol/mol), and high-purity (>99.996%) CF_4_ gas were used as the plasma precursor gas; they were purchased from Nanjing Special Gas Co., Ltd. (Nanjing, China). The used substrates were roving fabrics with the size of 15 mm × 15 mm × 1 mm supplied by Kunshan Zhangsheng Nano Technology Co., Ltd. (Kunshan, China). The weight of the fabrics with the scale of 10 × 10 mm^2^ was tested and the values were 77, 81, and 86 mg, respectively, and the average value was 81.3 mg with a standard deviation of 3.68. The used reagent including Ethylene glycol (AR, ≥99.7%) and acetone (AR, 99.2%) were obtained from Aladdin (Shanghai, China).

### 2.2. Plasma Deposition of Graphene, g-C_3_N_4_/Graphene and CF_2_ Modified g-C_3_N_4_/Graphene Films

The roving fabric substrate was washed with acetone solution for 20 min in an ultrasonic cleaning machine to wipe off the surface grease. Then, the fabric surface was washed repeatedly with alcohol and deionized water 3–4 times, and the fabric was dried at 80 °C for 60 min for subsequent experiments.

The synthesis scheme regarding the experimental procedures is shown in Figure 1. Plasma experiments were performed using a radio frequency (RF) plasma reactor (13.56 MHz) with capacitive coupling in pulsed mode. The substrates were put on the glass holder which is in the chamber center, and the plasma treatments were started after the ultimate chamber pressure reached the value of 3 × 10^−3^ Pa. In this study, different types of films can be obtained by changing the plasma gases, the graphene film was prepared by injecting the 0.25 mL/min plasma gas of Ar/CH_4_ at 10 Pa of stabilized working pressure for an exposure time of 10 min with RF power of 30 W. After the graphene film deposition, the Ar/CH_4_ gas was stopped, then the g-C_3_N_4_ groups were subsequently generated on the graphene by injecting 0.5 mL/min plasma gas of Ar/NH_3_ with the discharge power of 100 W at 24 Pa for 120 min, so the g-C_3_N_4_/graphene film was produced, and the sample was named as NG. Moreover, the g-C_3_N_4_/graphene film was further modified by the CF_4_ plasma which was generated at the RF power of 100 W under the working pressure of 24 Pa, and the CF_4_ gas of 0.5 mL/min was injected. Such modification process was performed for 120 min, finally, the CF_2_ modified g-C_3_N_4_/graphene film was prepared, and the sample was designed as FNG. The above abbreviations for the samples refer to the prepared film on the fabric.

### 2.3. Characterization Method

Nanostructures and morphologies of the films including graphene, NG, and FNG were tested using high-resolution transmission electron microscopy (HRTEM, Tecnai G2 F30 S-TWIN, FEI, Hillsboro, NH, USA). Due to the requirement of sample preparation for high-resolution transmission electron microscopy, the film should be stripped from the fabric and evenly dispersed in an alcohol solvent to characterize the morphology and structure of the film. The plasma-treated fabric was placed in an ultrasonic vibrator with an ethanol solution and vibrated for 1 h to ensure that the surface film of the fabric was fully dissolved in an alcohol solvent. The functional groups and chemical bonding types were tested by infrared spectroscopy (IR, Cary 640/670, Varian, Palo Alto, CA, USA) in ATR mode (ranging from 4000 to 400 cm^−1^) and X-ray photoelectron spectroscopy (XPS, ESCALAB 250Xi, ThermoFisher, Waltham, MA, USA) with the excitation source of a monochromatized Al Kα X-ray source (hν = 1486.68 eV). The binding energies of the XPS peaks were given with an accuracy of ±0.2 eV.

### 2.4. Contact Angle Measurements

Water contact angles (WCAs) of the fabrics covered by the produced films were measured by a contact angle analyzer (OCA, Dataphysics, Stuttgart, Germany), and the sessile drop method was used. Stability Testing was performed by an ultrasonic cleaning machine (JP-100, Jiemeng Cleaning Equipment Co., Ltd., Shenzhen, China), which provided an important indicator for preparing durable superhydrophobic fabric. The plasma-treated fabric was completely immersed in 95% alcohol and ultrasonic for 20 min, then removed and cleaned with deionized water three times. Finally, the plasma-treated fabric was placed in an oven and dried at 60 °C for 60 min. The contact angle of each sample after ultrasonic cleaning was secondly measured. The average angle was obtained by testing five different locations of each sample with ~10 μL of water droplets. Rolling angles (RAs) were detected using the sessile drop method and 10 μL deionized water droplets with the optical contact angle measuring instrument (OCA, Dataphysics, Stuttgart, Germany). RAs were determined by tilting the sample stage at a rate of one degree per second until the drops started moving and rolled off the sample surface.

### 2.5. Electrochemical Analysis

The electrochemical properties of the films were investigated by using the electrochemistry workstation (Interface 1010E type, Gamry, Philadelphia, PA, USA). The pure graphene, g-C_3_N_4_/graphene (NG), and CF_2_-modified g-C_3_N_4_/graphene (FNG) films were generated on the auxiliary electrode for measurements. The Electrochemical impedance spectroscopy (EIS) was obtained by testing the stable open circuit potential of the films immersed in 3.5% NaCl solution at frequencies ranging from 10^−2^–10^5^ Hz. Additionally, the Gamry Echem Analyst software was used to study the obtained PP and EIS curves.

### 2.6. Contact Angle Measurements

The first principles calculation of the obtained film was performed using the Dmol module in Materials Studio 7.0. The atomic and chemical combination was modified according to the characterization results based on the graphite model after unlocking its symmetry.

## 3. Results and Discussions

### 3.1. TEM Analysis on the Film Nanostructures

According to the PECVD deposition steps and different injected gases, three samples of the functional graphene films have been produced, which are the pure graphene, g-C_3_N_4_/graphene (NG), and CF_2_-modified g-C_3_N_4_/graphene (FNG) films. Firstly, the nanostructures of graphene and NG films were captured by TEM, as shown in Figure 2 and Figure 3. The obvious crystal structure occurred on the graphene film sample, and diffraction rings were (001) and (100) after the fast Fourier transform (FTT) as shown in Figure 2a. Zone A with an area of 5 × 5 nm was selected and operated under live FTT to obtain the auxiliary lattice image as shown in Figure 2b. The graphene grew with (001) and (100) advantages, which was consistent with the overall FTT. To be further analyzed, the orthogonal lattice fringes appeared on the surface of graphene, and the adjacent signal spacing was 7 pixels, corresponding to 1.43 μm converted length, meaning that the graphene quality was perfect, as shown in Figure 2c,d.

Then, the crystal structure of CF_2_ modified g-C_3_N_4_/graphene film (NG film) sample was detected, and the selected diffraction rings were (010) and (100) of g-C_3_N_4_ after the fast Fourier transform (FTT) as shown in Figure 3a. Zone B with an area of 5 × 5 nm was selected and operated under live FTT to obtain the auxiliary lattice image as shown in Figure 3b. The graphene grew to the direction of (010) and (100) advantages, which was consistent with the overall FTT. The orthogonal lattice fringes appeared on the surface of graphene, and the adjacent signal spacing was 12 pixels, corresponding to 2.45 μm converted length, as shown in Figure 3c,d.

For the sample of CF_2_-modified g-C_3_N_4_ film (FNG film), the nanostructure and EDS of FNG film were analyzed by TEM, as shown in Figure 4. The FNG film presented dendritic distribution and obvious fiber film structure [36], as shown in Figure 4a–c. After reduced FTT, it was found that the film presented a polycrystalline structure as a whole, as shown in Figure 4d. According to the color depth, two zones of A and B are selected for local analysis, and the results are shown in Figure 4e–h. Zone A presented polycrystalline diffraction, and the diffraction rings were (010), (110), and (120), respectively, which were mainly composed of graphene. The nano-structure of g-C_3_N_4_ was destroyed due to the modification of the F atom on the film surface, meanwhile, the CF_2_-modified g-C_3_N_4_ presented an amorphous structure, as shown in Zone B. The whole FNG film was scanned by EDS as shown in Figure 4i, and the result showed that the FNG film was mainly composed of C, F, and N elements, accompanied by a small amount of adsorbed oxygen.

The above TEM results reflected that all the prepared films are of good quality. For the NG film, g-C_3_N_4_ is grown in situ on the basis of graphene which presents a similar crystal structure to graphene, and the nanostructure of g-C_3_N_4_ is destroyed after an F atom was introduced on its surface. The FNG film exhibits an amorphous structure on the whole.

### 3.2. Chemical Structures by XPS and FTIR Analysis

The survey spectra of all the films were analyzed by XPS as shown in Figure 5a, and the fine spectra of C1s, N1s, and F1s in the films were shown in Figure 5b–d, whose atomic ratios were 34.7, 0.3, 4.6, and 60.4, respectively. The C1s spectrum was corrected for contaminated carbon at 284.5 eV. The chemical bonds of C-C, N-C=N, CF-CF_n_, CF_2,_ and CF_3_ appeared at the binding energy of 285.4, 288.4, 289.4, 291.1, and 292.9 eV, respectively [37,38]. The C-C bond originated from the graphene structure, the N-C=N bond originated from g-C_3_N_4_, and CF_x_ originated from the modified CF functional group. The N1s spectrum was fitted to C-NH_x_, N-(C)_3_, and N-C=N peaks, whose binding energy values were 401.6, 400.5, and 399.1 eV, respectively [39]. The main chemistry bond of the F1s spectrum was divided into CF_3_ and CF_2_ bonds, where the CF_3_ bond played the role of structure construction and the CF_2_ bond endowed the ability of superhydrophobic performance [40]. The valence band spectra of graphene, NG, and FNG films were tested during −2–4 eV binding energy as shown in Figure 5e, and the corresponding valence band value were 0.13, 1.22, and 1.54 eV, respectively, meaning that the NG and FNG enhanced the width of the valence band. The types and structures of functional groups on the surface of FNG film were studied by infrared spectroscopy, as shown in Figure 5f. The wavenumbers appeared at 1535 and 1699, corresponding to the C-N (I) and C-N (II) respiratory peaks, respectively [41].

From the above results, the composition and functional group types of the film can be obtained. The C-C bond, the N-C=N bond, and the CF_x_ bond are attributed to the graphene structure, g-C_3_N_4_, and modified functional group, respectively. The addition of nitrogen and fluorine atoms can enhance the valence bandwidth.

### 3.3. DFT Calculation

FNG film was analyzed based on density functional theory (DFT) as shown in Figure 6. N atoms replaced C atoms at the π end of graphene, and the g-C_3_N_4_ nanostructure with staggered C and N atoms was formed under the premise of ensuring the hexagonal “P1” spatial community of graphene [42]. At this time, the C-N bond length was 1.42 angms, and the angle of C-N-C and N-C-N was both 120°, as shown in Figure 6a. After introducing CF_4_ gas and continuous glow discharge, the C-N bond was broken, and an F atom was attached to the end of a C atom to form CF_2_. Such a process led to serious deformation of the g-C_3_N_4_ nanostructure, and C-N close to the F atom shrunk, while C-N far away from the F atom stretched [43], as shown in Figure 4b. The position of wave number within the range of 1620–1800 cm^−1^ was carefully scanned. It was found that the C-N absorption peaks can be fitted into three peaks, corresponding to C-N (compress), C-N (normal), and C-N (stretching) [44,45], as shown in Figure 6c, which was consistent with the simulation results in Figure 6b. More interestingly, the modified of CF_2_ functional groups increased the band gap of g-C_3_N_4_ from 0.022 eV to 0.217 eV, and the value of the valence band obtained from the density of state was 1.38 by studying the band gap of FNG film [45,46], which corresponded to the results in Figure 5e. There was a wide band gap on the path from F to Q, which was assigned to CF_2_ functional groups. This phenomenon caused a gap in the electrostatic potential on the graphene surface, which provided a channel for electron transfer, as shown in Figure 6d–f.

In combination with the results of infrared analysis, it can be concluded that the introduction of fluorine atoms led to severe deformation of the g-C_3_N_4_ nanostructure, which created gaps in the electrostatic potential of the graphene surface and provided channels for electron transport, which was consistent with the band gap results of XPS analysis.

### 3.4. Superhydrophobic Performance

The optical images of the contact angle were obtained by the contact angle measuring instrument, as shown in Figure 7. The fabric covered by the FNG film presented perfect superhydrophobic properties compared with the other two samples before and after ultrasonic cleaning, which relayed the superhydrophobic ability of functional groups. The surface of the Ar/CH_4_ preliminarily treated fabric substrate was hydrophilic with a static contact angle of 79.4°. After the second step of Ar/NH_3_ treatment, the surface of the treated fabric substrate transformed from a hydrophilic state to a hydrophobic state, with an increased static contact angle of 131.3°. Finally, after CF_4_ plasma treatment, the fabric substrate possessed superhydrophobic properties with a static contact angle of 156.2°. After ultrasonic cleaning for 20 min, the plasma-treated fabrics attached by the FNG, and NG film maintained favorable hydrophobic performance. Particularly, the superhydrophobic performance of the fabric attached by FNG film was stable on the whole. After ultrasonic cleaning, the fabric attached by FNG film still had a water contact angle as high as 147.6°, which was 8.6° different from the contact angle before ultrasonic cleaning, which indicated that the film exhibited good bonding force with the substrate, and the film exhibited good interlayer bonding force with the film. As for the rolling angles of the samples, the plasma-treated fabrics that were attached by the three kinds of films had similar rolling angles, and the sample with static contact angles over 150° showed a rolling angle of 35.8°.

The superhydrophobic performance of the prepared composite films in this work is compared with the experimental results of other authors, it can be seen in Figure 8 that the prepared CF_2_ modified g-C_3_N_4_/graphene composite film has a better water contact angle relative to the recent literature review.

### 3.5. Electrochemical Performance

#### 3.5.1. Polarization Curve Analysis

The electron transfer of the obtained film was established by an electrochemical workstation, and the polarization curves of the three films and the electrode are shown in Figure 9a. The Tafel extrapolation method was used to obtain the polarization current [51,52], which was further calculated to obtain polarization resistance, and the related results are shown in Table 1. The relationship between reactive oxygen species and polarization current was obtained by a first-order derivative of the anode part, as shown in Figure 9b. The polarization current of the electrode, graphene, NG, and FNG film was 7.031 × 10^−9^, 8.492 × 10^−9^, 7.603 × 10^−9^, and 1.059 × 10^−8^, respectively, which indicated that FNG film still had the ability to decompose reactive oxygen species at a higher polarization current and had good stability [53]. The FNG film had the highest polarization resistance, indicating a higher impedance of free electrons passing through the FNG film. Relevant scholars believed that the expression for polarization resistance was R_p_ = R_ct_ + W (where R_ct_ is the charge transfer resistance and W is the infinite diffusion resistance) [54,55], meaning that it was necessary to further study the state of charge transfer through electrochemical impedance spectroscopy (EIS).

#### 3.5.2. Electrochemical Impedance Spectroscopy Analysis

The prepared film was analyzed by electrochemical impedance spectroscopy as shown in Figure 10. The Nyquist curves of graphene, NG, and FNG films are shown in Figure 10a. The FNG film had similar electrochemical characteristics to graphene in the low-frequency region, while the radius of curvature of the polarization curve in the high-frequency region decreased, meaning that the FNG film had better electron transfer performance than graphene and NG [56,57]. Figure 10b shows the relationship between the frequency and phase angle of the graphene, NG, and FNG. The maximum phase angle corresponding to the response constant of FNG film appeared in the low-frequency region, while graphene and g-C_3_N_4_ appeared in the high-frequency region, indicating that FNG film had the characteristics of high pass and the ability of electron transfer [58,59]. Figure 10c shows the relationship between the frequency and impedance of graphene, NG, and FNG films. The impedance curves of the three films were consistent in the low-frequency region, and the FNG film had the highest impedance value in the high-frequency region. The FNG film was analyzed by electrochemical impedance spectroscopy [60,61], and the results are shown in Figure 10d,e, where the Rs is the resistance of NaCl solution, the equivalent circuit of the electrode is CPE_e_ (R_e_W), where CPE_e_ is the constant phase angle element of the electrode, R_e_ is the electron transfer resistance of the electrode, and W is the infinite diffusion resistance of graphene, NG, and the FNG film. The C_f_ and CPE_f_ are capacitance and constant phase angle elements of the film, respectively, R_po_ is pore resistance, CPE_dl_ is long phase angle element equivalent to multiple double capacitors, and R_ct_ is electron transfer resistance of the film. The data fitted by electrochemical impedance spectroscopy are shown in Table 1. The results showed that the graphene surface presented a capacitive state because the graphene only had six electron transition positions at the Dirac cone of Bruouin, which resulted in the overall capacitive state on the surface of the film [62,63]. The g-C_3_N_4_ was alternately arranged by C and N atoms, which formed an uneven three-dimensional structure [64]. The surface capacitance changed to a constant phase angle element, and the resistance of electron movement was reduced by 2% and 76.7%, respectively, compared with graphene and NG, which was consistent with the results of the first principles calculation.

## 4. Conclusions

In summary, by tuning the plasma deposition steps and injecting different plasma gases, the pure graphene, g-C_3_N_4_/graphene, and CF_2_-modified g-C_3_N_4_/graphene films were synthesized by PECVD. For the g-C_3_N_4_/graphene film, g-C_3_N_4_ was in-situ grown on the basis of graphene which possessed a similar crystal structure as graphene, while the CF_2_-modified g-C_3_N_4_/graphene film presented an amorphous structure on the whole. Based on density functional theory, first principles calculations combined with infrared experiments revealed that the introduction of fluorine atoms caused serious deformation of the g-C_3_N_4_ nanostructure, which created gaps in the electrostatic potential on the graphene surface and provided a channel for electron transfer. The CF_2_ functional groups promoted the treated fabric to exhibit excellent superhydrophobic properties and stability. The resistance of electron movement of CF_2_-modified g-C_3_N_4_/graphene film was reduced by 2% and 76.7%, respectively, compared with graphene and g-C_3_N_4_/graphene. Such sudden reduction of electron motion resistance resulted from longitudinal penetrating equipotential surface caused by introducing F atoms. This method provides ideas for expanding the application scope of multifunctional graphene, but some further studies still need to be conducted to face the problems of film thickness uniformity and large-area deposition for real applications.

## Figures and Tables

**Figure 1 nanomaterials-12-04387-f001:**
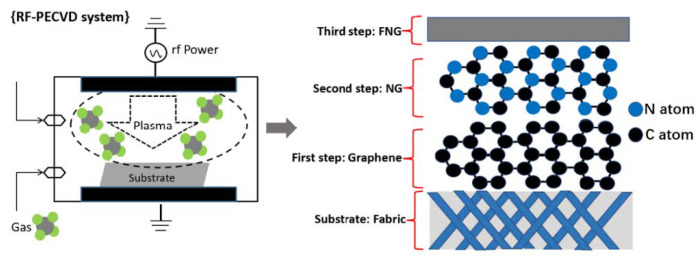
Schematic of graphene, NG and FNG films synthesized by PECVD technology.

**Figure 2 nanomaterials-12-04387-f002:**
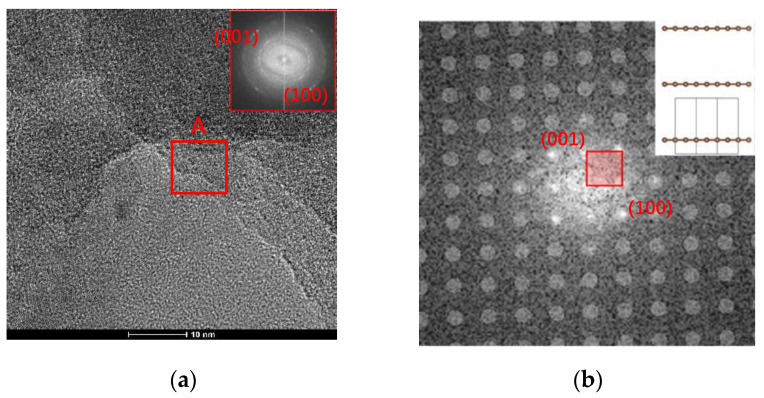
TEM observations on the graphene film nanostructures: (**a**) TEM image, (**b**) FTT image, (**c**) lattice stripe, (**d**) lattice spacing of graphene.

**Figure 3 nanomaterials-12-04387-f003:**
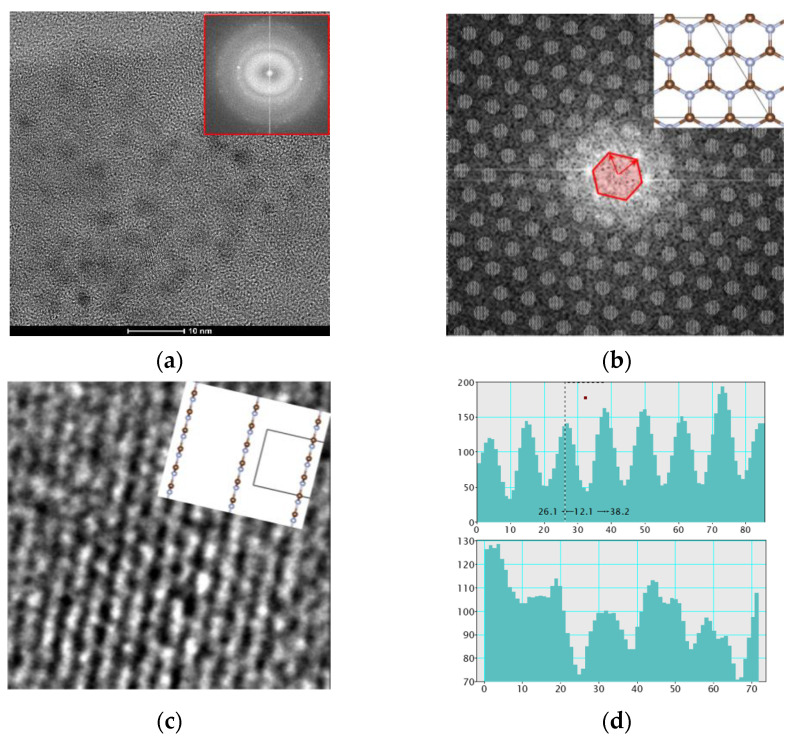
TEM observations on the NG film nanostructures: (**a**) TEM image, (**b**) FTT image, (**c**) lattice stripe, (**d**) lattice spacing of g-C_3_N_4_.

**Figure 4 nanomaterials-12-04387-f004:**
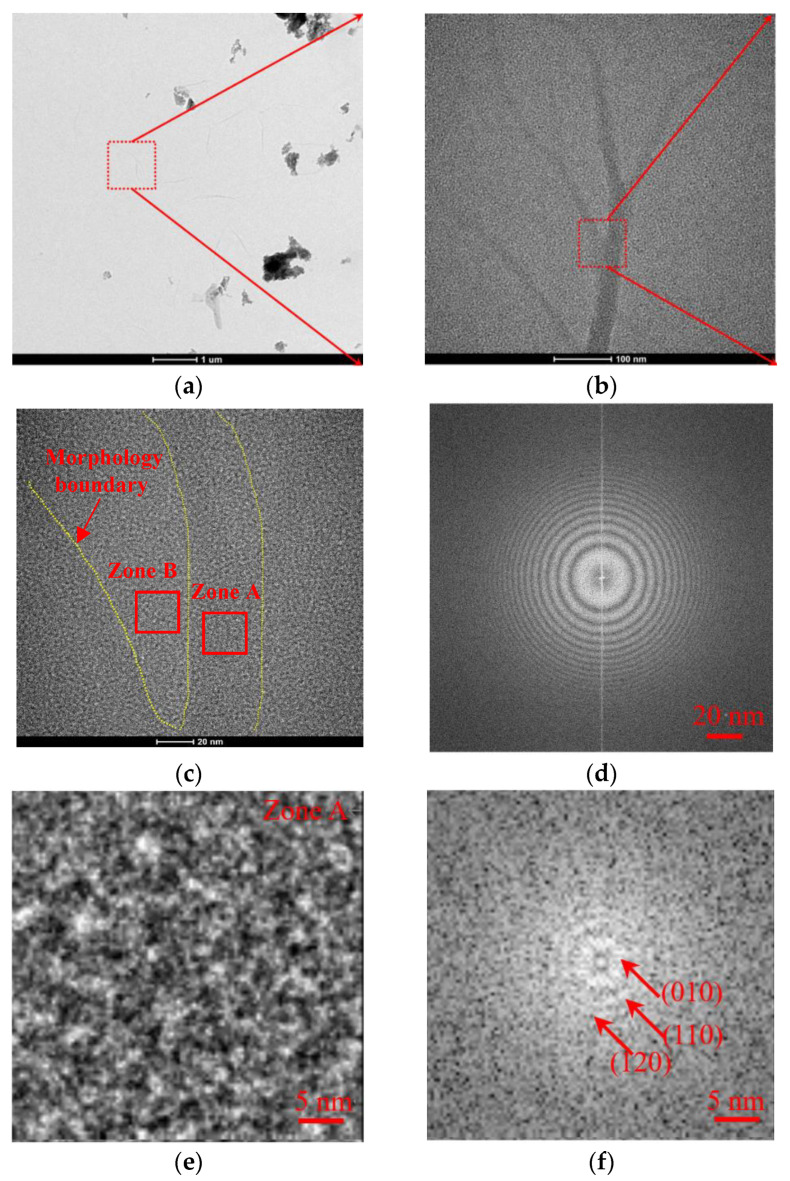
Morphology of FNG film under micron-scale (**a**), under submicron-scale (**b**), nano-scale (**c**), and reduced FFT (**d**). The nanostructure (**e**) and FFT (**f**) of zone A, and the amorphous structure (**g**) and FFT (**h**) of zone B were analyzed. Element distribution of FNG film was detected (**i**).

**Figure 5 nanomaterials-12-04387-f005:**
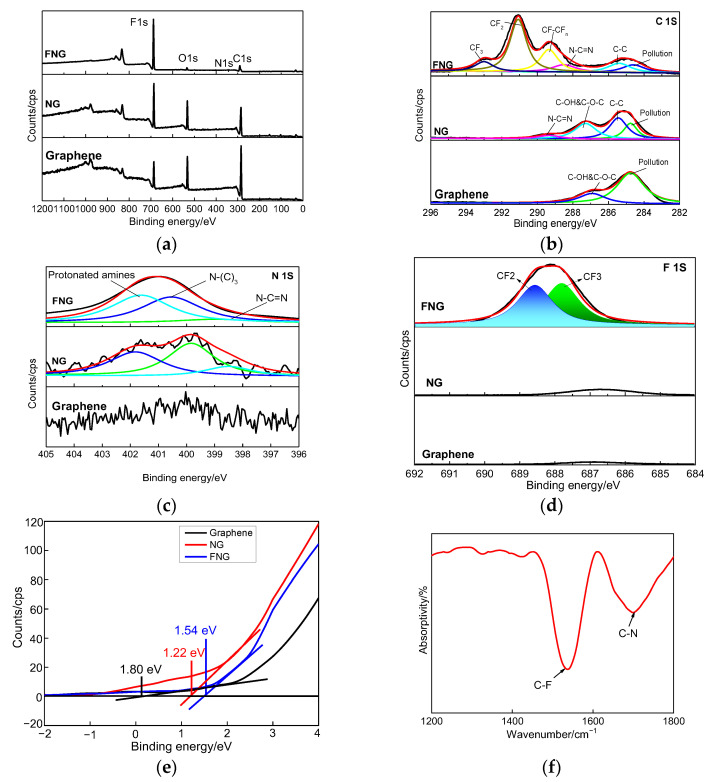
XPS and FTIR analysis of the obtained films: (**a**) XPS survey spectra, (**b**) C1s spectra, (**c**) N1s spectra, (**d**) F1s spectra, (**e**) valence band spectrum, (**f**) FTIR spectra.

**Figure 6 nanomaterials-12-04387-f006:**
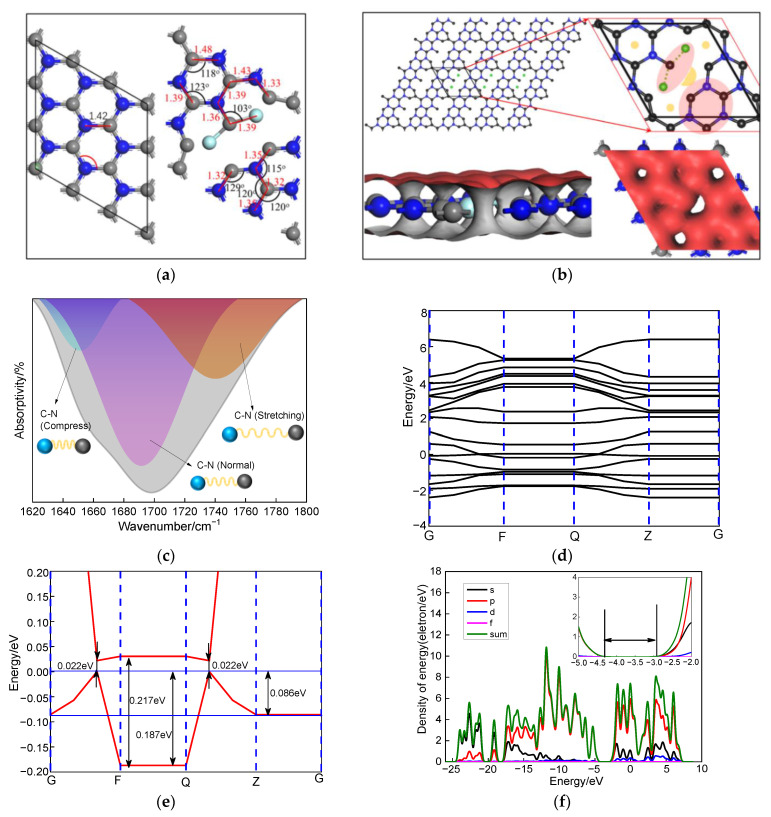
DFT calculation of FNG film including the (**a**) atomic structure of g-C_3_N_4_ and FNG film, (**b**) electron density of FNG film, (**c**) fine scanning of infrared spectrum, (**d**) band gap of FNG film, (**e**) band gap at Fermi of FNG film, (**f**) density of energy of FNG film.

**Figure 7 nanomaterials-12-04387-f007:**
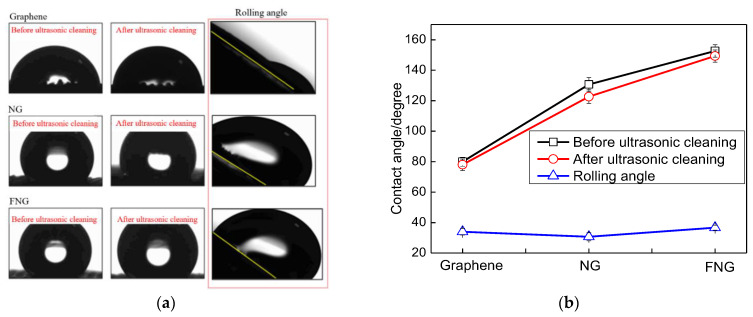
Contact angle measurements: (**a**) optical images of static state water droplets, (**b**) static state WCAs and Rolling state WCAs.

**Figure 8 nanomaterials-12-04387-f008:**
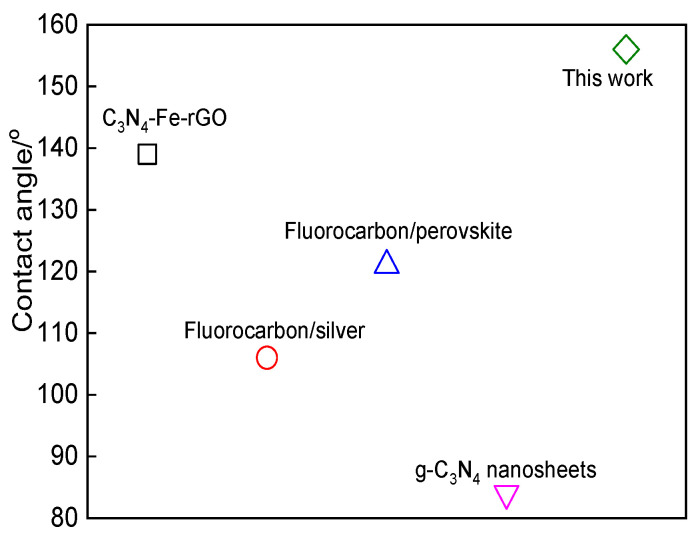
The recent literature review on g-C_3_N_4_ nanosheets [47], fluorocarbon/silver [48], fluorocarbon/perovskite [49], C_3_N_4_-Fe-rGO [50] and related water contact angle of superhydrophobic performance.

**Figure 9 nanomaterials-12-04387-f009:**
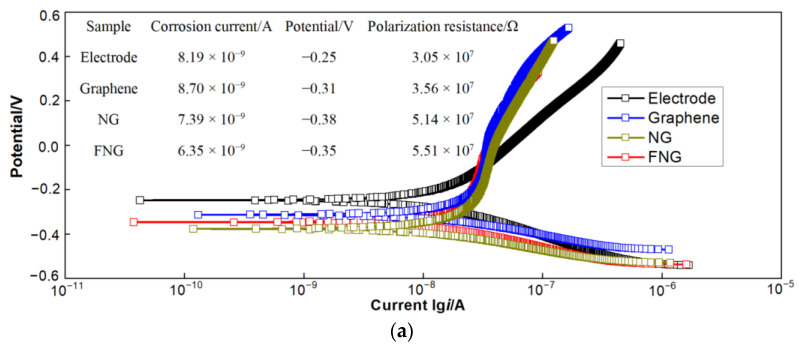
Dynamic polarization (**a**) and response of reactive oxygen curves (**b**) of electrode, graphene, NG, and FNG films.

**Figure 10 nanomaterials-12-04387-f010:**
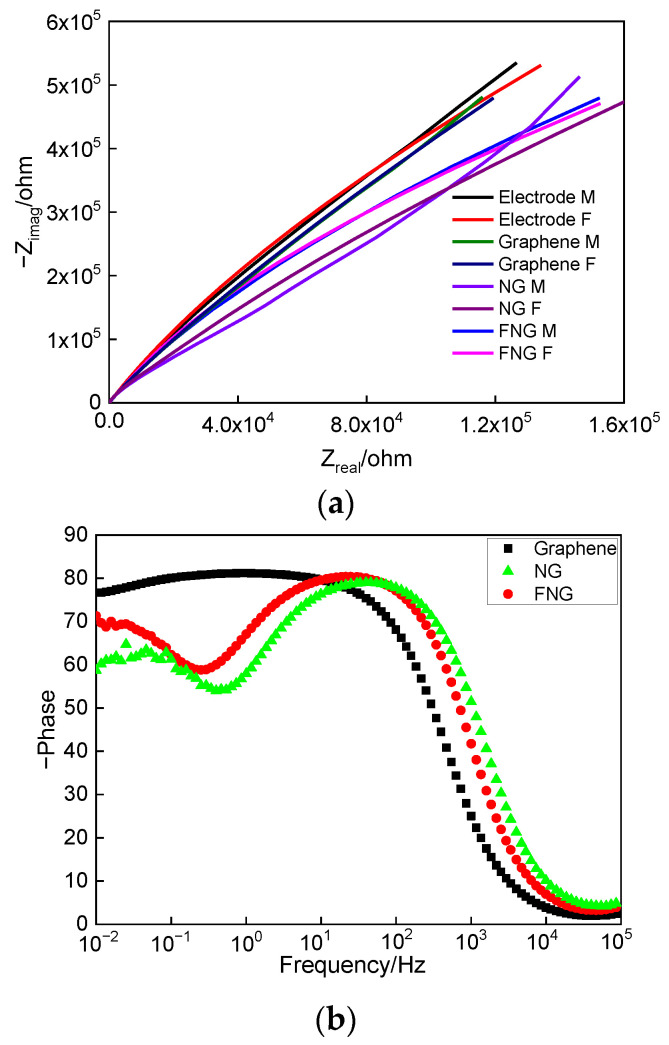
Nyquist curves (**a**), the relationship between frequency and phase angle (**b**), the relationship between frequency and impedance (**c**), and fitting circuit model (**d**,**e**) of graphene, NG, and FNG films.

**Table 1 nanomaterials-12-04387-t001:** Experimental parameters of EIS fitting.

Parameter	Electrode	Graphene	NG	FNG
Solution resistance (R_s_)/Ω	3.861 × 10	3.382 × 10	6.131	1.022
CPE_e_, Yo (S × sec^n^)	1.861 × 10^−5^	2.055 × 10^−5^	1.689 × 10^−5^	2.013 × 10^−5^
Freq power (*n*)/0 < *n* < 1	0.905	0.925	0.946	0.922
Electrode resistance Re/Ω	3.866 × 10^6^	6.381 × 10^5^	1.766 × 10^4^	1.506 × 10^6^
Warburg resistance/Ω	2.236 × 10^−6^	1.363 × 10^−6^	2.600 × 10^−6^	3.455 × 10^−6^
CPE_f_, Yo (S × sec^n^)			7.691 × 10^−9^	4.238 × 10^−8^
Freq power (*n*)/0 < *n* < 1			1	0.859
Capacitance F		7.758 × 10^−5^		
Pore resistance R_po_/Ω		9.926 × 10	1.430 × 10	2.99 × 10
CPE_dl_, Yo (S × sec^n^)		0.215 × 10^−3^	0.128 × 10^−3^	0.270 × 10^−4^
Freq power (*n*)/0 < *n* < 1		0.565	0.926	0.893
Charge transfer resistance R_ct_/Ω		1.809 × 10^4^	7.584 × 10^3^	1.768 × 10^3^
χ^2^ (10^−3^)	8.800 × 10^−2^	2.486 × 10^−2^	3.524 × 10^−1^	3.280 × 10^−2^
Solution resistance (R_s_)/Ω	3.861 × 10	3.382 × 10	6.131	1.022
CPE_e_, Yo (S × sec^n^)	1.861 × 10^−5^	2.055 × 10^−5^	1.689 × 10^−5^	2.013 × 10^−5^
Freq power (*n*)/0 < *n* < 1	0.905	0.925	0.946	0.922
Electrode resistance Re/Ω	3.866 × 10^6^	6.381 × 10^5^	1.766 × 10^4^	1.506 × 10^6^
Warburg resistance/Ω	2.236 × 10^−6^	1.363 × 10^−6^	2.600 × 10^−6^	3.455 × 10^−6^
CPE_f_, Yo (S × sec^n^)			7.691 × 10^−9^	4.238 × 10^−8^
Freq power (*n*)/0 < *n* < 1			1	0.859
Capacitance F		7.758 × 10^−5^		
Pore resistance R_po_/Ω		9.926 × 10	1.430 × 10	2.99 × 10
CPE_dl_, Yo (S × sec^n^)		0.215 × 10^−3^	0.128 × 10^−3^	0.270 × 10^−4^
Freq power (*n*)/0 < *n* < 1		0.565	0.926	0.893
Charge transfer resistance R_ct_/Ω		1.809 × 10^4^	7.584 × 10^3^	1.768 × 10^3^

## Data Availability

Not applicable.

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
