# Peer review of "Superhydrophobic and Electrochemical Performance of CF2-Modified g-C3N4/Graphene Composite Film Deposited by PECVD"

_nanomaterials, 2022, doi:10.3390/nano12244387_

Round 1
Reviewer 1 Report (Previous Reviewer 3)
The authors have replied to the reviewer's questions. I recommend the publication in this form.
Author Response
"The authors have replied to the reviewer's questions. I recommend the publication in this form."
Thanks very much for your comments and suggestions.
Reviewer 2 Report (New Reviewer)
Review report:
Authors reported “Superhydrophobic and electrochemical performance of CF2 modified g-C3N4/graphene composite film deposited by PECVD”. The organization of this work is good, and the discussion is well organized. The characterization and calculation are both solid for the conclusion. Nevertheless, I have some comments which are listed below.
1. The synthesis scheme should be included regarding the experimental procedures, and it should be revised in a detailed way.
2. The author claimed that their work is a novel investigation, however, myriads of works related to “CF2 modified g-C3N4/graphene“ have been published to date. So, authors should change the way of the presentation focusing on novelty. The introduction should be improved with a paragraph describing the novelty and importance of the work.
3. The authors must carefully claim their novelty in the INTRODUCTION. In addition, the authors need to do some formatting errors that should be carefully checked and corrected in the text.
4. The source and purity of all chemicals used should be specified. Authors should be looked at into below suggested references and can cite and take references regarding the “Source and Purity issues”: “Nanomaterials, 2022, 12(22), 3982” “Nanomaterials, 12 (2022), pp. 2330”, which references should be cited in your revised manuscript for better understanding.
5. A summary of key improvements compared to findings in the literature [provide a couple of references to indicate key improvements].
6. The surface area measurements are very important for active electrodes in supercapacitor applications. Please provide nitrogen adsorption and desorption (BET analysis) of the active materials (CF2 modified g-C3N4/graphene). Authors should be looked at into below suggested references with your work and cited them in your revised manuscript: “Electrochim. Acta, 330 (2019), Article 135261”, which given references should be cited in the revised manuscript.
7. The authors have missed their mass loadings of “CF2 modified g-C3N4/graphene” electrode materials should be included in the revised manuscript.
8. The selected area electron diffraction (SAED) is very crucial in supercapacitor applications. Please provide SAED of the active materials (CF2 modified g-C3N4/graphene).
9. Please provide the comparison table, which you need to prove that your material is superior to previously reported literature.
10. The reviewer also suggests that authors get professional English services to correct the grammatical error and refine the expressions in the body of the manuscript.
11. Authors should be trimmed/condensed the ‘Abstract’ and ‘Conclusion’ sections in the revised manuscript. Please keep highlights of the whole manuscript in both sections.

Author Response
Please see the attachment.

Reviewer 3 Report (New Reviewer)
In this work, the author reported: " Superhydrophobic and electrochemical performance of CF2-modified g-C3N4/graphene composite film deposited by PECVD" and systematically characterized using various physiochemical techniques. I strongly believe that the research part has been experimented with in a proper procedure. Therefore, I recommended this work for publication in the nanomaterial journal. However, some of the minor concerns should be addressed before proceeding with further actions.
1. Authors should include some results obtained from the characterization data.
2. Abstract requires more technical achievements from the proposed work to highlight the novelty of the work.
3. Keywords should not identical to title words. Should revise it.
4. The authors should clearly explain the innovation, research gap, market gap, market demand, and importance of their work in the manuscript's introduction. They should justify the value of the work and compare their work with previously similar published papers.
5. The authors did not explain the limitations in the introduction part. Should I assume that there are no limitations? It would be nice if they said the future perspectives and their limitations in the conclusion, which can attract more readers.
6. The section title 2.5 should be electrochemical test or electrochemical analysis.
7. Fig.3. SAED pattern is not clear enough.
8. Fig. 4, the X-axis title of FTIR spectrum should be wavenumber.
9. The typos and grammatical errors are scattered throughout the paper and need to be corrected with the utmost care.
Round 2
Reviewer 2 Report (New Reviewer)
It can be accepted in its current format
This manuscript is a resubmission of an earlier submission. The following is a list of the peer review reports and author responses from that submission.
Round 1
Reviewer 1 Report
This work presents very interesting results for functional graphene.However this work must be improved.
1) XPS. The interlretation of the xps results is somehow misleading. First, the positions for all your peaks are very strange! If you wish, I can provide the list of references that should be used while calibrating the BE scale and identifying peaks. Indeed, the setting C-NHx to 401.5 eV is radiculous. This is pisition for protonated amines. Also the c-H is 285 not 285.4 eV, nitride is not 288.4 eV. Please check it and need I can suggest the refs. Secondly, can you explain why the peajs looking as NO2 were visible in N1s spectrum? Third, Too many fluorinated groups. How is it possible? CF4 is not the best polymerizing gas. All these pudhes to the idea of wrong BE scale calubration.
2) FTIR. Results are strange. Your peaks somehow wrong. OH is a broad band
3) Impedance. You Nyquist plot is quite strange. All plots are very similar while he equivalent cicuit models are so different. This is not convincing. I cannot see the features of Warburg or CPE. Your plots somehow misleading.
Reviewer 2 Report
In my opinion, the manuscript needs to be extensively modified. There are some points that should be elucidated or improved and the study requires a major revision before publication.
- The key issue is to define the synthesized and characterized samples. The Materials and Methods part does not provide essential data to understand the study. It was mentioned that the authors applied the fabric as a substrate without any additional data: what kind of fabric and details of the synthesis process. Why authors selected fabric as a substrate with regards to textile structure roughness and non-uniform surface? The first paragraph of the Results and Discussion part provides only a superficial description of the synthesis without mentioning fabric use. A comprehensive description should be provided Materials and Methods part, also the scheme of the sample/synthesis process will be helpful. The abbreviations of the sample NG, FNG, and one for the graphene should be applied in the place where they appeared for the first time, with the synthesis process details. The samples and their abbreviations should be elucidated: graphene on fabric or graphene (if pure graphene, the way of removing it from the fabric should be given).
- The preparation of the electrode for electrochemical studies should be described in detail. What does it mean that “graphene film was generated on the auxiliary electrode” – synthesized or attached with the fabric (substrate) to the electrode and which way?
- What is the novelty of the study? The applied modification was reported previously to obtain superhydrophobic properties.
Reviewer 3 Report
The reading and understanding of the paper is not easy, therefore I cannot recommend the publication of the paper in this form. I think the authors should improve the following points:
English must certainly be improved. Not only in terms of grammar but also lexicon.
The data are presented but not discussed.
On how many samples was this study repeated? It is not acceptable if one graphene, one NG and one FNG were compared.
Standard deviations need to be shown in graph 6b and Table 1, reporting the averages over several measurements on several repetitions of the same sample.
Round 2
Reviewer 1 Report
The authors have performed a substantial revision and all questions were addressed. This paper can be now accepted
Reviewer 2 Report
I am afraid that is still difficult to understand the synthesis process. The authors did not implement necessary data on fabric characteristics: material (PET was mentioned in the response letter), the thickness of the fabric, mass per unit area, or linear mass of weft and warp yarns). In the Figure 1a (Responses, Fig.1 Micromorphology of the roving fabric and equipment) it is visible that the fabric is very loose, with large spaces between fibers, how that kind of substrate allowed to form uniform graphene films and how they look removed from the fabric? If the substrate is not sufficiently specified, we cannot determine its influence on the properties of the films (e.g. water contact angle) and their quality.
The description of the electrode is still the same (films were generated). The authors did not explain (Materials and Method) how the films were transferred from the fabric to the surface of the electrode and in which form.
Reviewer 3 Report
The authors have replied to the reviewer queries by showing the results coming out of three repetitions of the experiments. The results clearly show differences and the authors have declared they have chosen the "most reasonable". This seems to me result picking which is an unethical practice in science. Unless the authors are ready to explain which reasoning lead to chose one result over another I am afraid the paper cannot be published. Another way of facing this shortcoming would be to repeat the measurements several times until it is statistically clear that some results are "more reasonable" of others.